# A Hierarchical Theory for the Tensile Stiffness of Non-Buckling Fractal-Inspired Interconnects

**DOI:** 10.3390/nano13182542

**Published:** 2023-09-11

**Authors:** Yongkang Wang, Zanxin Zhou, Rui Li, Jianru Wang, Baolin Sha, Shuang Li, Yewang Su

**Affiliations:** 1State Key Laboratory of Nonlinear Mechanics, Institute of Mechanics, Chinese Academy of Sciences, Beijing 100190, China; 2School of Engineering Science, University of Chinese Academy of Sciences, Beijing 100049, China; 3State Key Laboratory of Structural Analysis, Optimization and CAE Software for Industrial Equipment, Department of Engineering Mechanics, International Research Center for Computational Mechanics, Dalian University of Technology, Dalian 116024, China; 4Xi’an Aerospace Propulsion Technology Institute, Xi’an 710025, China; 5The 41st Institute of the Fourth Academy of CASC, Xi’an 710025, China; 6Beijing Key Laboratory of Engineered Construction and Mechanobiology, Institute of Mechanics, Chinese Academy of Sciences, Beijing 100190, China

**Keywords:** fractal design, tensile stiffness, hierarchical theory, stretchable inorganic electronics

## Abstract

The design of non-buckling interconnects with thick sections has gained important applications in stretchable inorganic electronics due to their simultaneous achievement of high stretchability, low resistance, and low heat generation. However, at the same time, such a design sharply increased the tensile stiffness, which is detrimental to the conformal fit and skin comfort. Introducing the fractal design into the non-buckling interconnects is a promising approach to greatly reduce the tensile stiffness while maintaining other excellent performances. Here, a hierarchical theory is proposed for the tensile stiffness of the non-buckling fractal-inspired interconnects with an arbitrary shape at each order, which is verified by the finite element analysis. The results show that the tensile stiffness of the non-buckling fractal-inspired interconnects decreases with the increase in either the height/span ratio or the number of fractal orders but is not highly correlated with the ratio of the two adjacent dimensions. When the ratio of the two adjacent dimensions and height/span ratio are fixed, the tensile stiffness of the serpentine fractal-inspired interconnect is smaller than that of sinusoidal and zigzag fractal-inspired interconnects. These findings are of great significance for the design of non-buckling fractal-inspired interconnects of stretchable inorganic electronics.

## 1. Introduction

Stretchable inorganic electronics have been extensively developed over almost two decades for diverse applications, including health monitoring [1,2,3], medical treatment [4,5,6], human–computer interactions [7], aerospace engineering [8], etc. One of the most important technological innovations of stretchable inorganic electronics is the achievement of stretchability through mechanically guided structural designs, which guarantees that the electronic systems can conformally wrap arbitrarily complex target surfaces and maintain electronic functions. Among various strategies to achieve stretchability, the “island-bridge” structure is the most popular one [9,10]. The interconnects are referred to as the “bridges” and provide both stretchability and electrical conductivity, while the functional components reside at the “islands” and undergo negligible deformation during the stretch of the whole structure. Therefore, the structural design of the interconnects is crucial for stretchable inorganic electronics.

Up to now, various geometric layouts include the wavy structure [11,12,13], arc-shaped interconnects [9,14,15,16], serpentine interconnects [3,10,17,18,19,20,21], 2D spiral interconnects [22,23,24,25], 3D helical interconnects [26,27,28,29], etc., have been proposed successively. Corresponding analytical models are established to guide the optimization design of these structures. For example, Khang et al. provided an analytical solution for the wave structure [11], Lu et al. obtained the stretchability and compliance of freestanding serpentine interconnects [18,21], Pan et al. analyzed the serpentine interconnects on ultrathin elastomers [17], Li et al. proposed an analytical model for the buckled serpentine interconnects after the prestrain release in the substrate [30], and Ranji et al. discussed the stiffness of different types of folded-beam springs, including the serpentine spring, the U-shape spring, and crab-length beams [31]. To overcome the disadvantages of relatively low stretchability, high resistance, and high heat generation of the early buckling interconnects with thin sections, the non-buckling interconnects with thick sections have been proposed [10] and have gained important applications in stretchable inorganic electronics [8]. However, at the same time, such a non-buckling design sharply increased the tensile stiffness, which is detrimental to the conformal fit and skin comfort. Introducing the fractal-inspired design, which has been applied to the buckling interconnects for a high filling ratio and high stretchability (Figure 1 [6,32,33,34,35,36,37,38,39]), into the non-buckling interconnects is a promising approach to greatly reduce the tensile stiffness while maintaining the other excellent performances.

Mechanical analyses of fractal-inspired interconnects have made great progress in the past few years. Zhang et al. [40] have established an analytical model to evaluate the stretchability of the fractal-inspired structure and derived the recursive formulae at different orders of self-similarity (all orders have a similar shape) for the rectangular and serpentine interconnects. On the basis of mechanical optimization, interconnects with fractal-inspired design can provide a stretchability as high as ~90% with a surface-filling ratio of 70%. In addition, a hierarchical computational model [41] has been established to analyze the equivalent flexibilities and elastic stretchability of the interconnects with fractal-inspired designs. This hierarchical model extremely reduces the computational efforts of the full model without the loss of prediction accuracy. Furthermore, the elastic stiffness of the fractal-inspired interconnects has been analytically determined and verified by the experiments and the finite element analysis (FEA) [42]. Fan et al. [32] summarized six different patterns of metal interconnects. The elastic and plastic mechanics of these fractal-inspired layouts have been fully identified by both high-precision measurements and FEA, and it has also been pointed out that an increase in the angle of the arc section can improve the elastic stretchability of the horseshoe interconnects. More recently, the horseshoe fractal-inspired interconnects have been systemically investigated by Ma et al. through both the FEA and the experiments [43]. The nonlinear stress–strain curves of the horseshoe fractal-inspired interconnects have been discussed to achieve maximum elastic stretchability. Despite these important developments, the curve configurations of each order may not always remain similar in practical applications, which has not been studied by researchers.

In this paper, the tensile stiffness of the non-buckling fractal-inspired interconnects with arbitrary shapes at each order is investigated based on the energy method for the guidance of the structure design. In Section 2, a hierarchical theory is developed to obtain the load–displacement relationship of the interconnects, i.e., the flexibility matrix. Then the zigzag interconnects of multiple orders are studied in Section 3, and the relationships between the shape parameters and the deformation are particularly concerned and discussed. In Section 4, the hierarchical theory is verified by the FEA, and two key shape parameters for the multi-order fractal-inspired interconnects with different shapes are discussed. Concluding remarks are given in Section 5.

## 2. Theory of the Non-Buckling Fractal-Inspired Interconnects with Arbitrary Shape at Each Order

The non-buckling fractal-inspired interconnects at each order may not maintain the same pattern in practical applications, and it poses a challenge to the evaluation of the mechanical properties of the interconnects with a multi-order structure. This section aims to obtain the mechanical properties of the non-buckling fractal-inspired interconnects with arbitrary shapes at different orders, as shown in Figure 2. Due to the arbitrary shape of curves at different orders, one cannot simply obtain the relation between the flexibility of adjacent orders through the recursion formula. We first take the order-1 curve as an example to illustrate the theory as in Section 2.1, and then generalize the theoretical framework to an arbitrary order based on a concept of elastic strain energy density in Section 2.2, without the need for disassembly and force analysis from scratch as in previous studies [40,43]. After that, both the flexibility matrix and the tension stiffness of the entire interconnects, i.e., the order-*n* structure, are derived in Section 2.3.

### 2.1. Elastic Strain Energy Density of the First-Order Interconnects

This subsection introduces the non-buckling fractal-inspired interconnects that contain the arbitrary shape at each order, which is different from the previously proposed self-similar interconnects [40]. Figure 3a shows a representative order-1 cell, and its geometry is described by the local Cartesian coordinates (*Y*_1_, *Z*_1_)
(1)$$Y_{1}=Y_{1}\left(S_{1}\right),\;\;Z_{1}=Z_{1}\left(S_{1}\right),$$
where the origin of (*Y*_1_, *Z*_1_) is at the center of the order-1 structure; the curvilinear coordinate *S*_1_ is along the arc length of the order-1 structure, with *S*_1_ = 0 at the center. At two ends of the representative cell, we have $Y_{1}\left(-S_{1t}/2\right)=-Y_{1}\left(-S_{1t}/2\right)=0$, where $S_{1t}$ is the total length of the representative cell in the order-1 structure.

The interconnects at each order are modeled as beams. Figure 3b shows the sign convention for the positive internal axial force *P_n_*, shear force *Q_n_,* and bending moment *M_n_* in the order-*n* interconnects. Thus, for the order-1 structure, i.e., *n* = 1, let *P*_1_ and *Q*_1_ denote the internal forces along the local coordinates *Y*_1_ and *Z*_1_, respectively, at the center *S*_1_ = 0, and *M*_1_ is the bending moment for the order-1 structure. Then the axial and shear forces and bending moment at any curvilinear coordinate *S*_1_ are obtained from the force equilibrium as
(2)$$\begin{array}{l} M_{0}\left(S_{1}\right)=M_{1}+P_{1}Y_{1}\left(S_{1}\right)-Q_{1}Z_{1}\left(S_{1}\right)\\ P_{0}\left(S_{1}\right)=P_{1}\frac{dZ_{1}\left(S_{1}\right)}{dS_{1}}+Q_{1}\frac{dY_{1}\left(S_{1}\right)}{dS_{1}}\\ Q_{0}\left(S_{1}\right)=Q_{1}\frac{dZ_{1}\left(S_{1}\right)}{dS_{1}}-P_{1}\frac{dY_{1}\left(S_{1}\right)}{dS_{1}} \end{array},$$
where $M_{0}\left(S_{1}\right)$ is in a counterclockwise direction, $P_{0}\left(S_{1}\right)$ is along the tangent of *S*_1_, and $Q_{0}\left(S_{1}\right)$ is along the normal direction of *S*_1_ (rotating π/2 counterclockwise from the tangent of *S*_1_). Since the membrane energy is negligible as compared to the bending energy, the energy density of the order-1 structure is dominated by the bending energy per apparent length, which is given by
(3)$$U_{aver,1}\left(M_{1},P_{1},Q_{1}\right)=\frac{1}{2\bar{EI}L_{1}}\int_{0}^{S_{1t}}{\left[M_{0}\left(S_{1}\right)\right]}^{2}dS_{1}.$$

The nominal energy density is defined by the total elastic energy divided by the apparent length of the representative cell $L_{1}$ (see Figure 3a). $\bar{EI}$ is the bending stiffness of the order-1 structure. Substitution of Equation (2) into Equation (3) gives the energy density
(4)$$U_{aver,1}\left(M_{1},P_{1},Q_{1}\right)=\frac{1}{2\bar{EI}}\left(\alpha_{1,1}M_{1}^{2}+\alpha_{1,2}P_{1}^{2}L_{1}^{2}+\alpha_{1,3}Q_{1}^{2}L_{1}^{2}+\alpha_{1,4}M_{1}P_{1}L_{1}+\alpha_{1,5}M_{1}Q_{1}L_{1}+\alpha_{1,6}P_{1}Q_{1}L_{1}^{2}\right),$$
where
(5)$$\alpha_{1,1}=\gamma_{1},\;\alpha_{1,2}=\gamma_{2},\;\alpha_{1,3}=\gamma_{3},\;\alpha_{1,4}=2\gamma_{4},\;\alpha_{1,5}=-2\gamma_{5},\;\alpha_{1,6}=-2\gamma_{6}.$$

Here, $\alpha_{n,m}$ are the shape parameters only related to the geometry of interconnects, regardless of the overall size, and the subscripts *n* and *m* represent the order of the interconnects and the parameter number. For the order-1 structure, i.e., *n* = 1, the parameters in Equation (5) are given by
(6)$$\begin{array}{l} \gamma_{1}=\frac{S_{1t}}{L_{1}},\;\gamma_{2}=\frac{1}{L_{1}^{3}}\int_{0}^{S_{1t}}{\left[Y_{1}\left(S_{1}\right)\right]}^{2}dS_{1},\;\gamma_{3}=\frac{1}{L_{1}^{3}}\int_{0}^{S_{1t}}{\left[Z_{1}\left(S_{1}\right)\right]}^{2}dS_{1}\\ \gamma_{4}=\frac{1}{L_{1}^{2}}\int_{0}^{S_{1y}}Y_{1}\left(S_{1}\right)dS_{1},\;\gamma_{5}=\frac{1}{L_{1}^{2}}\int_{0}^{S_{1t}}Z_{1}\left(S_{1}\right)dS_{1},\;\gamma_{6}=\frac{1}{L_{1}^{3}}\int_{0}^{S_{1t}}Y_{1}\left(S_{1}\right)Z_{1}\left(S_{1}\right)dS_{1} \end{array}.$$

### 2.2. Recursion Relations between the Interconnects of Adjacent Two Orders

As shown in Figure 3b, the order-*n* structure is an arbitrary shape. Similar to the analysis of the order-1 structure, the geometric shape of the representative order-*n* unit cell can be described by the local Cartesian coordinates (*Y_n_*, *Z_n_*)
(7)$$Y_{n}=Y_{n}\left(S_{n}\right),\;\;Z_{n}=Z_{n}\left(S_{n}\right).$$

Here, *S_n_* is the curvilinear coordinate along the arc length of the order-*n* structure. Let *P_n_* and *Q_n_* denote the internal forces along the local coordinates and *M_n_* bending moment at *S_n_* = 0. The bending moment and the internal forces in the order-*n* structure can be generally given as
(8)$$\begin{array}{l} M_{n-1}\left(S_{n}\right)=M_{n}+P_{n}Y_{n}\left(S_{n}\right)-Q_{n}Z_{n}\left(S_{n}\right)\\ P_{n-1}\left(S_{n}\right)=P_{n}\frac{dZ_{n}\left(S_{n}\right)}{dS_{n}}+Q_{n}\frac{dY_{n}\left(S_{n}\right)}{dS_{n}}\\ Q_{n-1}\left(S_{n}\right)=Q_{n}\frac{dZ_{n}\left(S_{n}\right)}{dS_{n}}-P_{n}\frac{dY_{n}\left(S_{n}\right)}{dS_{n}} \end{array},$$
where the signs of $M_{n-1}\left(S_{n}\right)$, $P_{n-1}\left(S_{n}\right)$, and $Q_{n-1}\left(S_{n}\right)$ are defined to be similar to those in the order-1 structure. Once the internal forces and the moment are obtained in the order (*n* − 1) by the derivation from the higher order, the energy density for order *n* is the integration of that of the order *n* − 1,
(9)$$U_{aver,n}\left(M_{n},P_{n},Q_{n}\right)=\frac{1}{L_{n}}\int_{0}^{S_{nt}}U_{aver,n-1}\left(M_{n-1},P_{n-1},Q_{n-1}\right)dS_{n},$$
where $S_{nt}$ and $L_{n}$ are the total arc length and the apparent length of the representative cell in the order-*n* interconnects, as shown in Figure 3b. In the case of linear elastic deformation, the energy density of the order-(*n* − 1) interconnects can be expressed as a quadratic form of the generalized forces,
(10)$$U_{aver,n-1}\left(M_{n-1},P_{n-1},Q_{n-1}\right)=\frac{1}{2\bar{EI}}\left(\begin{array}{l} \alpha_{n-1,1}M_{n-1}^{2}+\alpha_{n-1,2}P_{n-1}^{2}L_{n-1}^{2}+\alpha_{n-1,3}Q_{n-1}^{2}L_{n-1}^{2}\\ +\alpha_{n-1,4}M_{n-1}P_{n-1}L_{n-1}+\alpha_{n-1,5}M_{n-1}Q_{n-1}L_{n-1}+\alpha_{n-1,6}P_{n-1}Q_{n-1}L_{n-1}^{2} \end{array}\right).$$

Then, by combining Equations (8)–(10), the energy density of the order-*n* interconnects can be calculated by the integration of that of the order *n* − 1, which can also be expressed as a quadratic form,
(11)$$U_{aver,n}\left(M_{n},P_{n},Q_{n}\right)=\frac{1}{2\bar{EI}}\left(\begin{array}{l} \alpha_{n,1}M_{n}^{2}+\alpha_{n,2}P_{n}^{2}L_{n}^{2}+\alpha_{n,3}Q_{n}^{2}L_{n}^{2}\\ +\alpha_{n,4}M_{n}P_{n}L_{n}+\alpha_{n,5}M_{n}Q_{n}L_{n}+\alpha_{n,6}P_{n}Q_{n}L_{n}^{2} \end{array}\right).$$

Here, the parameters $\alpha_{n,m}$ (*m* = 1~6) in Equation (11) can be given by



(12)
$$\begin{array}{l} \alpha_{n,1}=\alpha_{n-1,1}\beta_{1}\\ \alpha_{n,2}=\frac{1}{\eta_{n,n-1}^{2}}\left(\eta_{n,n-1}^{2}\alpha_{n-1,1}\beta_{2}+\alpha_{n-1,2}\beta_{8}+\alpha_{n-1,3}\beta_{7}+\eta_{n,n-1}\alpha_{n-1,4}\beta_{12}-\eta_{n,n-1}\alpha_{n-1,5}\beta_{10}-\alpha_{n-1,6}\beta_{9}\right)\\ \alpha_{n,3}=\frac{1}{\eta_{n,n-1}^{2}}\left(\eta_{n,n-1}^{2}\alpha_{n-1,1}\beta_{3}+\alpha_{n-1,2}\beta_{7}+\alpha_{n-1,3}\beta_{8}-\eta_{n,n-1}\alpha_{n-1,4}\beta_{13}-\eta_{n,n-1}\alpha_{n-1,5}\beta_{11}+\alpha_{n-1,6}\beta_{9}\right)\\ \alpha_{n,4}=\frac{1}{\eta_{n,n-1}}\left(2\eta_{n,n-1}\alpha_{n-1,1}\beta_{4}+\alpha_{n-1,4}\beta_{15}-\alpha_{n-1,5}\beta_{14}\right)\\ \alpha_{n,5}=\frac{1}{\eta_{n,n-1}}\left(-2\eta_{n,n-1}\alpha_{n-1,1}\beta_{5}+\alpha_{n-1,4}\beta_{14}+\alpha_{n-1,5}\beta_{15}\right)\\ \alpha_{n,6}=\frac{1}{\eta_{n,n-1}^{2}}\left[\begin{array}{l} -2\eta_{n,n-1}^{2}\alpha_{n-1,1}\beta_{6}+2\left(\alpha_{n-1,2}-\alpha_{n-1,3}\right)\beta_{9}+\eta_{n,n-1}\alpha_{n-1,4}\left(\beta_{10}-\beta_{11}\right)\\ +\eta_{n,n-1}\alpha_{n-1,5}\left(\beta_{12}+\beta_{13}\right)+\alpha_{n-1,6}\left(\beta_{8}-\beta_{7}\right) \end{array}\right] \end{array}.$$



In the recursive relationship above, $\eta_{n,n-1}=L_{n}/L_{n-1}$ is the ratio of the two adjacent dimensions. The parameters $\beta_{i}$ (*i* = 1~15) are only related to the geometric shape of the structure at each order, given by



(13)
$$\begin{array}{l} \beta_{1}=\frac{S_{nt}}{L_{n}},\;\beta_{2}=\frac{1}{L_{n}^{3}}\int_{0}^{S_{nt}}{\left[Y_{n}\left(S_{n}\right)\right]}^{2}dS_{n},\;\beta_{3}=\frac{1}{L_{n}^{3}}\int_{0}^{S_{nt}}{\left[Z_{n}\left(S_{n}\right)\right]}^{2}dS_{n}\\ \beta_{4}=\frac{1}{L_{n}^{2}}\int_{0}^{S_{ny}}Y_{n}\left(S_{n}\right)dS_{n},\;\beta_{5}=\frac{1}{L_{n}^{2}}\int_{0}^{S_{nt}}Z_{n}\left(S_{n}\right)dS_{n},\;\beta_{6}=\frac{1}{L_{n}^{3}}\int_{0}^{S_{nt}}Y_{n}\left(S_{n}\right)Z_{n}\left(S_{n}\right)dS_{n}\\ \beta_{7}=\frac{1}{L_{n}}\int_{0}^{S_{nt}}{\left[\frac{dY_{n}\left(S_{n}\right)}{dS_{n}}\right]}^{2}dS_{n},\;\beta_{8}=\frac{1}{L_{n}}\int_{0}^{S_{nt}}{\left[\frac{dZ_{n}\left(S_{n}\right)}{dS_{n}}\right]}^{2}dS_{n},\;\beta_{9}=\frac{1}{L_{n}}\int_{0}^{S_{nt}}\frac{dY_{n}\left(S_{n}\right)}{dS_{n}}\frac{dZ_{n}\left(S_{n}\right)}{dS_{n}}dS_{n}\\ \beta_{10}=\frac{1}{L_{n}^{2}}\int_{0}^{S_{nt}}Y_{n}\left(S_{n}\right)\frac{dY_{n}\left(S_{n}\right)}{dS_{n}}dS_{n},\;\beta_{11}=\frac{1}{L_{n}^{2}}\int_{0}^{S_{nt}}Z_{n}\left(S_{n}\right)\frac{dZ_{n}\left(S_{n}\right)}{dS_{n}}dS_{n}\\ \beta_{12}=\frac{1}{L_{n}^{2}}\int_{0}^{S_{nt}}Y_{n}\left(S_{n}\right)\frac{dZ_{n}\left(S_{n}\right)}{dS_{n}}dS_{n},\;\beta_{13}=\frac{1}{L_{n}^{2}}\int_{0}^{S_{nt}}Z_{n}\left(S_{n}\right)\frac{dY_{n}\left(S_{n}\right)}{dS_{n}}dS_{n}\\ \beta_{14}=\frac{1}{L_{n}}\int_{0}^{S_{nt}}\frac{dY_{n}\left(S_{n}\right)}{dS_{n}}dS_{n},\;\beta_{15}=\frac{1}{L_{n}}\int_{0}^{S_{nt}}\frac{dZ_{n}\left(S_{n}\right)}{dS_{n}}dS_{n} \end{array}.$$



From the derivation above, it can be further confirmed that the shape parameters $\alpha_{n,m}$ are only related to the geometry of interconnects at each order. The relationship between the shape parameters of the structure of the adjacent two orders is given by Equation (12), which can be further written in a matrix form as
(14)$$\left[\begin{array}{c} \alpha_{n,1}\\ \alpha_{n,2}\\ \alpha_{n,3}\\ \alpha_{n,4}\\ \alpha_{n,5}\\ \alpha_{n,6} \end{array}\right]={\left[D\right]}_{n,n-1}\left[\begin{array}{c} \alpha_{n-1,1}\\ \alpha_{n-1,2}\\ \alpha_{n-1,3}\\ \alpha_{n-1,4}\\ \alpha_{n-1,5}\\ \alpha_{n-1,6} \end{array}\right],$$
where the recursive coefficient matrix ${\left[D\right]}_{n,n-1}$ can be expressed as



(15)
$${\left[D\right]}_{n,n-1}=\left[\begin{array}{cccccc} \beta_{1} & 0 & 0 & 0 & 0 & 0\\ \beta_{2} & \frac{1}{\eta_{n,n-1}^{2}}\beta_{8} & \frac{1}{\eta_{n,n-1}^{2}}\beta_{7} & \frac{1}{\eta_{n,n-1}}\beta_{12} & -\frac{1}{\eta_{n,n-1}}\beta_{10} & -\frac{1}{\eta_{n,n-1}^{2}}\beta_{9}\\ \beta_{3} & \frac{1}{\eta_{n,n-1}^{2}}\beta_{7} & \frac{1}{\eta_{n,n-1}^{2}}\beta_{8} & -\frac{1}{\eta_{n,n-1}}\beta_{13} & -\frac{1}{\eta_{n,n-1}}\beta_{11} & \frac{1}{\eta_{n,n-1}^{2}}\beta_{9}\\ 2\beta_{4} & 0 & 0 & \frac{1}{\eta_{n,n-1}}\beta_{15} & -\frac{1}{\eta_{n,n-1}}\beta_{14} & 0\\ -2\beta_{5} & 0 & 0 & \frac{1}{\eta_{n,n-1}}\beta_{14} & \frac{1}{\eta_{n,n-1}}\beta_{15} & 0\\ -2\beta_{6} & \frac{2}{\eta_{n,n-1}^{2}}\beta_{9} & -\frac{2}{\eta_{n,n-1}^{2}}\beta_{9} & \frac{1}{\eta_{n,n-1}}\left(\beta_{10}-\beta_{11}\right) & \frac{1}{\eta_{n,n-1}}\left(\beta_{12}+\beta_{13}\right) & \frac{1}{\eta_{n,n-1}^{2}}\left(\beta_{8}-\beta_{7}\right) \end{array}\right].$$



Thus, the shape parameters $\alpha_{n,1\sim 6}$ in an arbitrary order can be derived from the order-1 structure,
(16)$$\left[\alpha_{n,1\sim 6}\right]=\prod_{j=2}^{n}{\left[D\right]}_{j,j-1}\left[\alpha_{1,1\sim 6}\right],(n>1)$$

Here, the initial shape parameters $\left[\alpha_{1,1\sim 6}\right]$ have been given in Equation (5).

### 2.3. Flexibility Matrix and Tensile Stiffness of Order-n Interconnects

In this section, the flexibility matrix of the order-*n* interconnects is established based on the recursion relations above. A representative cell of the order-*n* interconnects is illustrated in Figure 3c. Considering fixing the left end of the cell, the forces and bending moment, as well as the corresponding generalized displacements, are shown in Figure 3c. ${\tilde{u}}_{n}$ is the generalized displacement parallel to the axial direction, ${\tilde{v}}_{n}$ is the generalized displacement perpendicular to the axial direction, and ${\tilde{\theta }}_{n}$ is the rotation angle of the right end. The relationship between the displacements at the right end and the forces/moment can be written in the following general form:(17)$$\left[\begin{array}{c} {\tilde{\theta }}_{n}\\ {\tilde{u}}_{n}\\ {\tilde{v}}_{n} \end{array}\right]=\left[S_{n}\right]\left[\begin{array}{c} {\tilde{M}}_{n}\\ {\tilde{P}}_{n}\\ {\tilde{Q}}_{n} \end{array}\right],$$
where $\left[S_{n}\right]$ is the flexibility matrix. 

To obtain the flexibility matrix, the equivalent translational forces at the center point are first calculated from the external loading at the right end as
(18)$$\left[\begin{array}{c} M_{n}\\ P_{n}\\ Q_{n} \end{array}\right]=\left[\begin{array}{ccc} 1 & 0 & \frac{1}{2}L_{n}\\ 0 & 1 & 0\\ 0 & 0 & 1 \end{array}\right]\left[\begin{array}{c} {\tilde{M}}_{n}\\ {\tilde{P}}_{n}\\ {\tilde{Q}}_{n} \end{array}\right],$$

Then, according to Equation (11), the total deformation energy of the order-*n* interconnects is obtained as
(19)$$U_{n}\left(M_{n},P_{n},Q_{n}\right)=U_{aver,n}\left(M_{n},P_{n},Q_{n}\right)L_{n}=\frac{L_{n}}{2\bar{EI}}{\left[\begin{array}{c} M_{n}\\ P_{n}\\ Q_{n} \end{array}\right]}^{T}\left[\begin{array}{ccc} \alpha_{n,1} & \frac{1}{2}\alpha_{n,4}L_{n} & \frac{1}{2}\alpha_{n,5}L_{n}\\ \frac{1}{2}\alpha_{n,4}L_{n} & \alpha_{n,2}L_{n}^{2} & \frac{1}{2}\alpha_{n,6}L_{n}^{2}\\ \frac{1}{2}\alpha_{n,5}L_{n} & \frac{1}{2}\alpha_{n,6}L_{n}^{2} & \alpha_{n,3}L_{n}^{2} \end{array}\right]\left[\begin{array}{c} M_{n}\\ P_{n}\\ Q_{n} \end{array}\right].$$

Substitution of the equivalent translational forces in Equation (18) into Equation (19) gives a new formula of the total deformation energy in terms of the external loading at the left end:



(20)
$$\begin{array}{l} U_{n}\left(M_{n},P_{n},Q_{n}\right)=U_{aver,n}\left(M_{n},P_{n},Q_{n}\right)L_{n}=\\ =\frac{L_{n}}{2\bar{EI}}{\left[\begin{array}{c} {\tilde{M}}_{n}\\ {\tilde{P}}_{n}\\ {\tilde{Q}}_{n} \end{array}\right]}^{T}\left[\begin{array}{ccc} 1 & 0 & 0\\ 0 & 1 & 0\\ \frac{1}{2}L_{n} & 0 & 1 \end{array}\right]\left[\begin{array}{ccc} \alpha_{n,1} & \frac{1}{2}\alpha_{n,4}L_{n} & \frac{1}{2}\alpha_{n,5}L_{n}\\ \frac{1}{2}\alpha_{n,4}L_{n} & \alpha_{n,2}L_{n}^{2} & \frac{1}{2}\alpha_{n,6}L_{n}^{2}\\ \frac{1}{2}\alpha_{n,5}L_{n} & \frac{1}{2}\alpha_{n,6}L_{n}^{2} & \alpha_{n,3}L_{n}^{2} \end{array}\right]\left[\begin{array}{ccc} 1 & 0 & \frac{1}{2}L_{n}\\ 0 & 1 & 0\\ 0 & 0 & 1 \end{array}\right]\left[\begin{array}{c} {\tilde{M}}_{n}\\ {\tilde{P}}_{n}\\ {\tilde{Q}}_{n} \end{array}\right]\\ =\frac{L_{n}}{2\bar{EI}}{\left[\begin{array}{c} {\tilde{M}}_{n}\\ {\tilde{P}}_{n}\\ {\tilde{Q}}_{n} \end{array}\right]}^{T}\left[\begin{array}{ccc} \alpha_{n,1} & \frac{1}{2}\alpha_{n,4}L_{n} & \frac{1}{2}\left(\alpha_{n,1}+\alpha_{n,5}\right)L_{n}\\ \frac{1}{2}\alpha_{n,4}L_{n} & \alpha_{n,2}L_{n}^{2} & \frac{1}{2}\left(\alpha_{n,6}+\frac{1}{2}\alpha_{n,4}\right)L_{n}^{2}\\ \frac{1}{2}\left(\alpha_{n,1}+\alpha_{n,5}\right)L_{n} & \frac{1}{2}\left(\alpha_{n,6}+\frac{1}{2}\alpha_{n,4}\right)L_{n}^{2} & \left(\frac{1}{4}\alpha_{n,1}+\alpha_{n,3}+\frac{1}{2}\alpha_{n,5}\right)L_{n}^{2} \end{array}\right]\left[\begin{array}{c} {\tilde{M}}_{n}\\ {\tilde{P}}_{n}\\ {\tilde{Q}}_{n} \end{array}.\right] \end{array}$$



Thus, the flexibility matrix can be obtained as
(21)$$\left[S_{n}\right]=\frac{L_{n}}{\bar{EI}}\left[\begin{array}{ccc} \alpha_{n,1} & \frac{1}{2}\alpha_{n,4}L_{n} & \frac{1}{2}\left(\alpha_{n,1}+\alpha_{n,5}\right)L_{n}\\ \frac{1}{2}\alpha_{n,4}L_{n} & \alpha_{n,2}L_{n}^{2} & \frac{1}{2}\left(\alpha_{n,6}+\frac{1}{2}\alpha_{n,4}\right)L_{n}^{2}\\ \frac{1}{2}\left(\alpha_{n,1}+\alpha_{n,5}\right)L_{n} & \frac{1}{2}\left(\alpha_{n,6}+\frac{1}{2}\alpha_{n,4}\right)L_{n}^{2} & \left(\frac{1}{4}\alpha_{n,1}+\alpha_{n,3}+\frac{1}{2}\alpha_{n,5}\right)L_{n}^{2} \end{array}\right].$$

The dimensionless flexibility matrix is defined as
(22)$$\left[{\bar{S}}_{n}\right]=\left[\begin{array}{ccc} \alpha_{n,1} & \frac{1}{2}\alpha_{n,4} & \frac{1}{2}\left(\alpha_{n,1}+\alpha_{n,5}\right)\\ \frac{1}{2}\alpha_{n,4} & \alpha_{n,2} & \frac{1}{2}\left(\alpha_{n,6}+\frac{1}{2}\alpha_{n,4}\right)\\ \frac{1}{2}\left(\alpha_{n,1}+\alpha_{n,5}\right) & \frac{1}{2}\left(\alpha_{n,6}+\frac{1}{2}\alpha_{n,4}\right) & \left(\frac{1}{4}\alpha_{n,1}+\alpha_{n,3}+\frac{1}{2}\alpha_{n,5}\right) \end{array}\right].$$

The tensile stiffness can be further derived from the flexibility matrix. As shown in Figure 3d, we consider that the left side of the representative cell in the order-*n* interconnects is fixed and the right end can only move horizontally, i.e., ${\tilde{\theta }}_{n}=0$ and ${\tilde{v}}_{n}=0$, and then the relationship between the external forces and displacements at the end can be expressed as
(23)$$\left[\begin{array}{c} {\tilde{M}}_{n}\\ {\tilde{P}}_{n}\\ {\tilde{Q}}_{n} \end{array}\right]={\left[S_{n}\right]}^{-1}\left[\begin{array}{c} {\tilde{\theta }}_{n}\\ {\tilde{u}}_{n}\\ {\tilde{v}}_{n} \end{array}\right]=\frac{\bar{E}\bar{I}}{L_{n}{}^{2}}{\tilde{u}}_{n}\left[\begin{array}{c} k_{12}\\ \begin{array}{c} \frac{k_{22}}{L_{n}}\\ \frac{k_{32}}{L_{n}} \end{array} \end{array}\right],$$
where
(24)$$\begin{array}{l} k_{12}=\frac{2(\alpha_{n,1}+\alpha_{n,5})\alpha_{n,6}-(4\alpha_{n,3}+\alpha_{n,5})\alpha_{n,4}}{2k_{0}}\\ k_{22}=\frac{4\alpha_{n,1}\alpha_{n,3}-\alpha_{n,5}{}^{2}}{k_{0}}\\ k_{32}=\frac{\alpha_{n,4}\alpha_{n,5}-2\alpha_{n,1}\alpha_{n,6}}{k_{0}}\\ k_{0}=4\alpha_{n,1}\alpha_{n,2}\alpha_{n,3}+\alpha_{n,4}\alpha_{n,5}\alpha_{n,6}-\alpha_{n,1}\alpha_{n,6}{}^{2}-\alpha_{n,2}\alpha_{n,5}{}^{2}-\alpha_{n,3}\alpha_{n,4}{}^{2} \end{array}.$$

It can be obtained from Equation (23) that
(25)$${\tilde{P}}_{n}=K_{n}{\tilde{u}}_{n},$$
where $K_{n}$ is the tensile stiffness of the whole structure,
(26)$$K_{n}=\frac{\bar{EI}}{L_{n}^{3}}k_{22}=\frac{\bar{EI}}{L_{n}^{3}}\frac{4\alpha_{n,1}\alpha_{n,3}-\alpha_{n,5}^{2}}{4\alpha_{n,1}\alpha_{n,2}\alpha_{n,3}+\alpha_{n,4}\alpha_{n,5}\alpha_{n,6}-\alpha_{n,1}\alpha_{n,6}^{2}-\alpha_{n,2}\alpha_{n,5}^{2}-\alpha_{n,3}\alpha_{n,4}^{2}}.$$

## 3. Theoretical Analysis of Self-Similar Zigzag Interconnects

In Section 2, the flexibility matrix and the tensile stiffness of the order-*n* fractal-inspired interconnects with arbitrary shapes are obtained. Here, as a simple case, the analytical model for the self-similar zigzag fractal-inspired interconnects is established to show the practical operation process.

### 3.1. Shape Parameters and Recursive Formula

The geometrical parameters of the order-1 zigzag interconnects are illustrated in Figure 4a. Specifically, the included angle at the inflection point of the zigzag interconnects is 2φ. The geometric shape of the structure can be expressed as
(27)$$\begin{array}{l} Y_{1}=\left\{\begin{array}{l} \cot\varphi \left(\sin\varphi S_{1}-\frac{L_{1}}{2}\right)+\frac{\cot\varphi L_{1}}{2}\;\;(0\leq S_{1}\leq \frac{L_{1}}{4\sin\varphi })\\ -\cot\varphi \left(\sin\varphi S_{1}-\frac{L_{1}}{2}\right)\;\;\;\;\;\;\;\;\;\;\;\;\;(\frac{L_{1}}{4\sin\varphi }<S_{1}<\frac{3L_{1}}{4\sin\varphi })\\ \cot\varphi \left(\sin\varphi S_{1}-\frac{L_{1}}{2}\right)-\frac{\cot\varphi L_{1}}{2}\;\;(\frac{3L_{1}}{4\sin\varphi }\leq S_{1}\leq \frac{L_{1}}{\sin\varphi }) \end{array}\right.\\ Z_{1}=\sin\varphi S_{1}-\frac{L_{1}}{2} \end{array}.$$

Substitution of Equation (27) into Equation (13) gives the parameters of order-1 zigzag interconnects as 



(28)
$$\begin{array}{l} \beta_{1}=\frac{S_{1t}}{L_{1}}=csc\varphi ,\;\;\beta_{2}=\frac{1}{L_{1}^{3}}\int_{0}^{S_{1t}}{\left[Y_{1}\left(S_{1}\right)\right]}^{2}dS_{1}=\frac{1}{48}\cot^{2}\varphi \csc\varphi \\ \beta_{3}=\frac{1}{L_{1}^{3}}\int_{0}^{S_{1t}}{\left[Z_{1}\left(S_{1}\right)\right]}^{2}dS_{1}=\frac{1}{12}\csc\varphi ,\beta_{4}=\beta_{5}=0\\ \beta_{6}=\frac{1}{L_{1}^{3}}\int_{0}^{S_{1t}}Y_{1}\left(S_{1}\right)Z_{1}\left(S_{1}\right)dS_{1}=-\frac{1}{32}\cot\varphi \csc\varphi \\ \beta_{7}=\frac{1}{L_{1}}\int_{0}^{S_{1t}}{\left[\frac{dY_{1}\left(S_{1}\right)}{dS_{1}}\right]}^{2}dS_{1}=\cot\varphi \cos\varphi \\ \beta_{8}=\frac{1}{L_{1}}\int_{0}^{S_{1t}}{\left[\frac{dZ_{1}\left(S_{1}\right)}{dS_{1}}\right]}^{2}dS_{1}=\sin\varphi \\ \beta_{9}=\beta_{10}=\beta_{11}=\beta_{12}=\beta_{13}=\beta_{14}=0\\ \beta_{15}=\frac{1}{L_{1}}\int_{0}^{S_{1t}}\frac{dZ_{1}\left(S_{1}\right)}{dS_{1}}dS_{1}=1 \end{array}.$$



Such that the shape parameters in the order-1 zigzag interconnects can be expressed as
(29)$${\left[\alpha_{1,1\sim 6}\right]}^{T}=\left[\begin{array}{cccccc} \frac{1}{\sin\varphi } & \frac{1}{48}\frac{\cos^{2}\varphi }{\sin^{3}\varphi } & \frac{1}{12}\frac{1}{\sin\varphi } & 0 & 0 & \frac{1}{16}\frac{\cos\varphi }{\sin^{2}\varphi } \end{array}\right],$$

The recursive formula in the self-similar zigzag interconnects becomes
(30)$$\left[\alpha_{n,1\sim 6}\right]=\prod_{j=2}^{n}{\left[D\right]}_{j,j-1}\left[\alpha_{1,1\sim 6}\right],$$
where the recursive coefficient matrix is
(31)$${\left[D\right]}_{n,n-1}=\left[\begin{array}{cccccc} \csc\varphi & 0 & 0 & 0 & 0 & 0\\ \frac{1}{48}\cot^{2}\varphi \csc\varphi & \frac{1}{\eta_{n,n-1}^{2}}\sin\varphi & \frac{1}{\eta_{n,n-1}^{2}}\cot\varphi \cos\varphi & 0 & 0 & 0\\ \frac{1}{12}\csc\varphi & \frac{1}{\eta_{n,n-1}^{2}}\cot\varphi \cos\varphi & \frac{1}{\eta_{n,n-1}^{2}}\sin\varphi & 0 & 0 & 0\\ 0 & 0 & 0 & \frac{1}{\eta_{n,n-1}} & 0 & 0\\ 0 & 0 & 0 & 0 & \frac{1}{\eta_{n,n-1}} & 0\\ \frac{1}{16}\cot\varphi \csc\varphi & 0 & 0 & 0 & 0 & -\frac{1}{\eta_{n,n-1}^{2}}\csc\varphi \cos2\varphi \end{array}\right],$$

In practical applications, the apparent length of the adjacent representative cells differs greatly. For the case of $\eta_{n,n-1}=\eta >>1$ of the self-similar zigzag interconnects, the recursive coefficient matrix in Equation (31) can be simplified to
(32)$${\left[D\right]}_{n,n-1}=\left[\begin{array}{cccccc} \csc\varphi & 0 & 0 & 0 & 0 & 0\\ \frac{1}{48}\cot^{2}\varphi \csc\varphi & 0 & 0 & 0 & 0 & 0\\ \frac{1}{12}\csc\varphi & 0 & 0 & 0 & 0 & 0\\ 0 & 0 & 0 & 0 & 0 & 0\\ 0 & 0 & 0 & 0 & 0 & 0\\ \frac{1}{16}\cot\varphi \csc\varphi & 0 & 0 & 0 & 0 & 0 \end{array}\right].$$

Then the recursive formula becomes
(33)$$\left[\alpha_{n,1\sim 6}\right]=\csc^{n-1}\varphi \left[\alpha_{1,1\sim 6}\right].$$

### 3.2. Flexibility Matrix of the Self-Similar Zigzag Interconnects

The order-*n* zigzag interconnects are subjected to an axial force ${\tilde{P}}_{n}$, a shear force ${\tilde{Q}}_{n}$, and a bending moment ${\tilde{M}}_{n}$ at the right end, as shown in Figure 4b. The flexibility matrix can be given as the same as that in Equation (21). Each component in the flexibility matrix, i.e., the shape parameters, can be calculated by Equation (29). 

The flexibility matrix can be further simplified by considering two different combinations of external loads. (1) For ${\tilde{M}}_{n}=0,\;{\tilde{Q}}_{n}\;=0,\;{\tilde{P}}_{n}\;\neq 0$, Equations (17) and (21) give the rotation angle ${\tilde{\theta }}_{n}=\left[{\tilde{P}}_{n}L_{n}^{2}/\left(2\bar{EI}\right)\right]\alpha_{n,4}$, and under these external loads, ${\tilde{\theta }}_{n}=0$, it can be determined that $\alpha_{n,4}=0$. (2) For ${\tilde{P}}_{n}=0,{\tilde{Q}}_{n}\;\neq 0,\;{\tilde{M}}_{n}=-{\tilde{Q}}_{n}L_{n}/2$, Equations (17) and (21) give the rotation angle ${\tilde{\theta }}_{n}=\left[{\tilde{Q}}_{n}L_{n}^{2}/\left(2\bar{EI}\right)\right]\alpha_{n,5}$, and under these external loads, ${\tilde{\theta }}_{n}=0$, it can be determined that $\alpha_{n,5}=0$. Thus, the flexibility matrix of the order-*n* zigzag interconnects can then be expressed as
(34)$$\left[S_{n}\right]=\frac{L_{n}}{\bar{EI}}\left[\begin{array}{ccc} \alpha_{n,1} & 0 & \frac{1}{2}\alpha_{n,1}L_{n}\\ 0 & \alpha_{n,2}L_{n}^{2} & \frac{1}{2}\alpha_{n,6}L_{n}^{2}\\ \frac{1}{2}\alpha_{n,1}L_{n} & \frac{1}{2}\alpha_{n,6}L_{n}^{2} & \left(\frac{1}{4}\alpha_{n,1}+\alpha_{n,3}\right)L_{n}^{2} \end{array}\right].$$

The tensile stiffness $K_{n}$ of the zigzag interconnects can be calculated by Equation (26). 

## 4. Finite Element Analysis and Discussion on the Shape Parameters

In this section, a series of finite element analyses are carried out to verify the validity of the hierarchical theory. The “island-bridge” structure and corresponding geometric parameters are shown in Figure 5a. The space *L* between adjacent rigid islands is 4 mm, and the thickness and width of the interconnects are 0.2 μm and 1 μm, respectively. The elastic modulus *E* of the interconnects is 78 GPa. Two kinds of fractal-inspired interconnects with different shapes in each order are calculated and compared with the hierarchical theory. The dimensionless flexibility matrix $\left[{\bar{S}}_{n}\right]$ and the dimensional tensile stiffness ${\bar{K}}_{n}$ are discussed.

### 4.1. Dimensionless Flexibility Matrix

In this subsection, two different interconnects are investigated to clarify the hierarchical theory. As shown in Figure 5b, the order-1 curve is serpentine and the order-2 curve is sinusoidal (type I). The ratio of the horizontal length $L_{1}:L_{2}$ in the two orders is 1:10.6. Subsequently, the fractal structures with three orders also analyzed as the type II shown in Figure 5b, the order-1 curve is sinusoidal, and the order-2 curve and order-3 curve are both zigzag. The ratio of the horizontal length of each order $L_{1}:L_{2}:L_{3}$ is $1:8\sqrt{2}:64$. $S_{11}$ is the angular value generated by applying a unit bending moment at the right end; $S_{12}$ is the Z-direction displacement value generated by applying a unit bending moment at the right end; $S_{13}$ is the Y-direction displacement value generated by applying a unit bending moment at the right end; $S_{21}$ is the angular value generated by applying a unit force in the Z direction at the right end; $S_{22}$ is the Z-direction displacement value generated by applying a unit force in the Z direction at the right end; $S_{23}$ is the Y-direction displacement value generated by applying a unit force in the Z direction at the right end; $S_{31}$ is the angular value generated by applying a unit force in the Y direction at the right end; $S_{32}$ is the Z-direction displacement value generated by applying a unit force in the Y direction at the right end; $S_{33}$ is the Y-direction displacement value generated by applying a unit force in the Y direction at the right end. Both the analytical calculation and the FEA are performed to evaluate each component of the dimensionless flexibility matrix in Equation (22), as shown in Figure 5c,d. The analytical results match the FEA results well, which proves the validity of the hierarchical theory.

### 4.2. Effect of the Shape Parameters γ and η

Furthermore, two key geometrical parameters are investigated in the self-similar interconnects in this subsection. At each fractal order, the interconnects have the same, centrosymmetric shape, as shown in Figure 6a–c. The height and the span of the order-*n* structure are $H_{n}$ and $L_{n}$, respectively. Then the height/span ratio for different cells can be defined by $\gamma =H_{n}/L_{n}$. The ratio of the two adjacent dimensions $\eta_{n,n-1}$ is set to be a constant η. The dimensional tensile stiffness ${\bar{K}}_{n}$, which can be obtained from Equation (26) by ${\bar{K}}_{n}=K_{n}L_{n}{}^{3}/\bar{E}\bar{I}$, is evaluated by both analytical calculation and FEA. In detail, the ${\bar{K}}_{n}$ of the self-similar interconnects with different orders and height/span ratio are calculated as a function of η, as shown in Figure 6d. It can be seen that the hierarchical theory can predict the deformation behavior of the fractal interconnects with an arbitrary shape accurately. The self-similar structures become more flexible with the increase in the order. With the increase in the height/span ratio γ, the tensile stiffness also decreases. However, the parameter η has a very limited effect on the stiffness of the structure when it becomes larger.

For the ratio of the two adjacent dimensions η=6 and the height/span ratio γ = 0.5, the relationship between dimensionless tensile stiffness ${\bar{K}}_{n}$ and the number of fractal orders *n* for the zigzag, sinusoidal, and serpentine fractal-inspired interconnects was obtained through the hierarchical theory, as shown in Figure 6e. It can be seen that the dimensionless tensile stiffness ${\bar{K}}_{n}$ of the three types of interconnects decreases with the increase in the fractal orders. Under the same fractal orders, the dimensionless tensile stiffness of serpentine interconnects is the smallest, the dimensionless tensile stiffness of sinusoidal interconnects is in the middle, and the dimensionless tensile stiffness of zigzag interconnects is the highest. For the ratio of the two adjacent dimensions η=6 and the fractal order *n* = 1, the relationship between dimensionless tensile stiffness ${\bar{K}}_{n}$ and the height/span ratio γ for the zigzag, sinusoidal, and serpentine fractal-inspired interconnects was obtained through the hierarchical theory, as shown in Figure 6f. The dimensionless tensile stiffness ${\bar{K}}_{n}$ of the three types of interconnects decreases with the increase in the height/span ratio γ. When γ ≤ 1.6, the dimensionless tensile stiffness of the serpentine interconnects is the smallest, the dimensionless tensile stiffness of the sinusoidal interconnects is in the middle, and the dimensionless tensile stiffness of the zigzag interconnects is the highest; however, when γ > 1.6, the dimensionless tensile stiffness of the three interconnects is very close and tends toward zero. For the ratio of the two adjacent dimensions η=6 and orders *n* from 1 to 4, the relationship between dimensionless tensile stiffness ${\bar{K}}_{n}$ and the height/span ratio γ for the serpentine fractal-inspired interconnects was obtained through the hierarchical theory, as shown in Figure 6g. The dimensionless tensile stiffness ${\bar{K}}_{n}$ of the serpentine interconnects decreases with the increase in the height/span ratio γ. Under the same height/span ratio γ, the tensile stiffness also decreases with the increase in the fractal orders *n*.

## 5. Conclusions

This paper develops a hierarchical theory for the non-buckling fractal-inspired interconnects with arbitrary shapes in each order. The flexibility matrix and the tensile stiffness of the non-buckling fractal-inspired interconnects are derived based on the energy method by introducing a concept of elastic strain energy density, which is further verified by the FEA. Both the analytical calculation and the FEA are carried out to investigate the effect of the shape parameters on the dimensionless flexibility matrix $\left[{\bar{S}}_{n}\right]$ and the dimensional tensile stiffness ${\bar{K}}_{n}$. The results show that the tensile stiffness of the non-buckling fractal-inspired interconnects decreases with the increase in either the height/span ratio γ or the number of fractal orders, but is not highly correlated with the ratio of the two adjacent dimensions *η*. When the ratio of the two adjacent dimensions *η* and the height/span ratio γ are fixed, the tensile stiffness of the serpentine fractal-inspired interconnect is smaller than those of sinusoidal and zigzag fractal-inspired interconnects. These findings are of great significance for the design of the non-buckling fractal-inspired interconnects of stretchable inorganic electronics.

## Figures and Tables

**Figure 1 nanomaterials-13-02542-f001:**
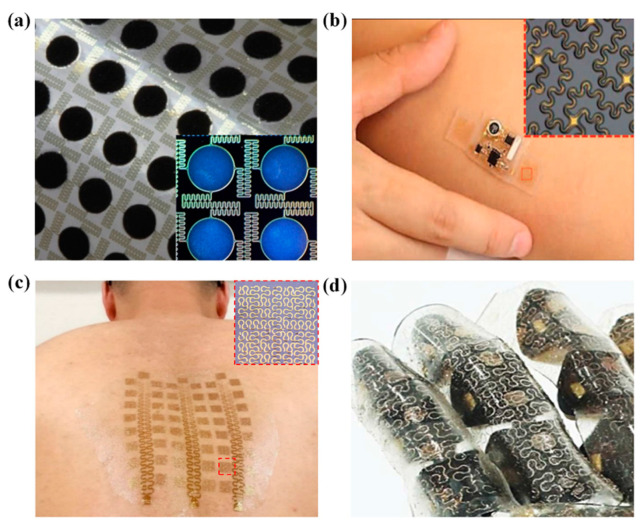
Typical applications of the fractal-inspired design in stretchable inorganic electronics: (**a**) Stretchable batteries [35]; (**b**) a wireless electrophysiological sensor [36]; (**c**) multifunctional and MRI-compatible epidermal electrical interfaces [37]; (**d**) smart prosthetic skin instrumented with various sensors and actuators [38].

**Figure 2 nanomaterials-13-02542-f002:**
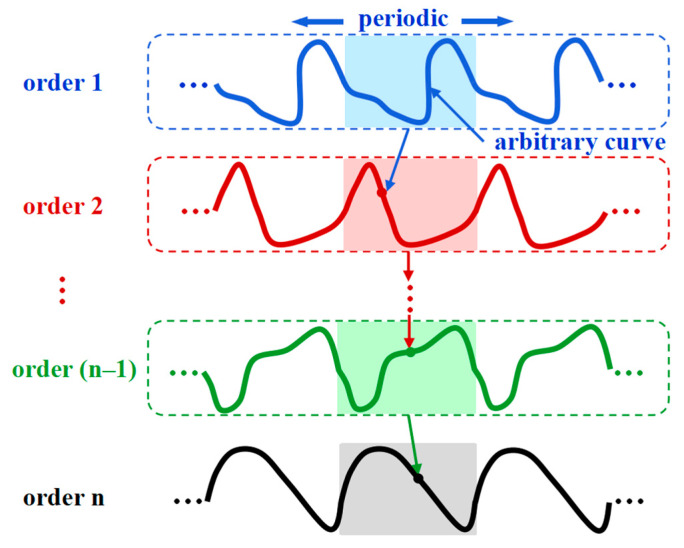
Schematic illustration of the geometric construction of the non-buckling fractal-inspired interconnects with arbitrary shapes.

**Figure 3 nanomaterials-13-02542-f003:**
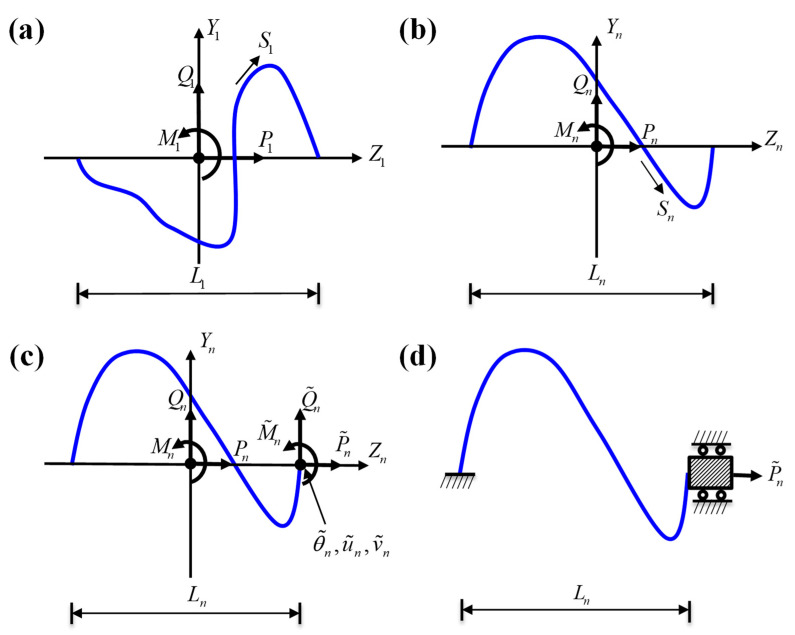
(**a**) The curvilinear coordinate *S*_1_ along the arc length and the internal forces *P*_1_, *Q*_1_, and *M*_1_ in the order-1 structure. (**b**) The curvilinear coordinate *S*_n_ and the internal forces in the order-*n* structure. (**c**) The internal forces and the corresponding generalized displacements in the order-*n* structure. (**d**) The simplified model of the order-*n* structure fixed at the left end and sliding at the right end.

**Figure 4 nanomaterials-13-02542-f004:**
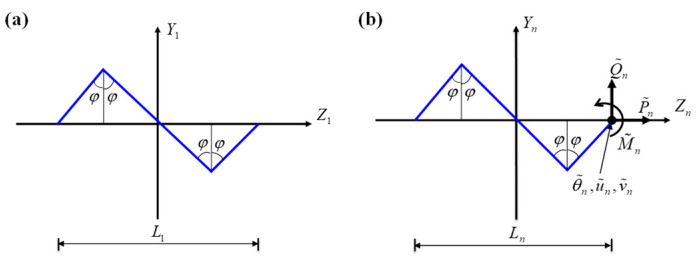
(**a**) The geometric construction of the order-1 zigzag interconnects. (**b**) The internal forces and generalized displacements in the order-*n* zigzag interconnects.

**Figure 5 nanomaterials-13-02542-f005:**
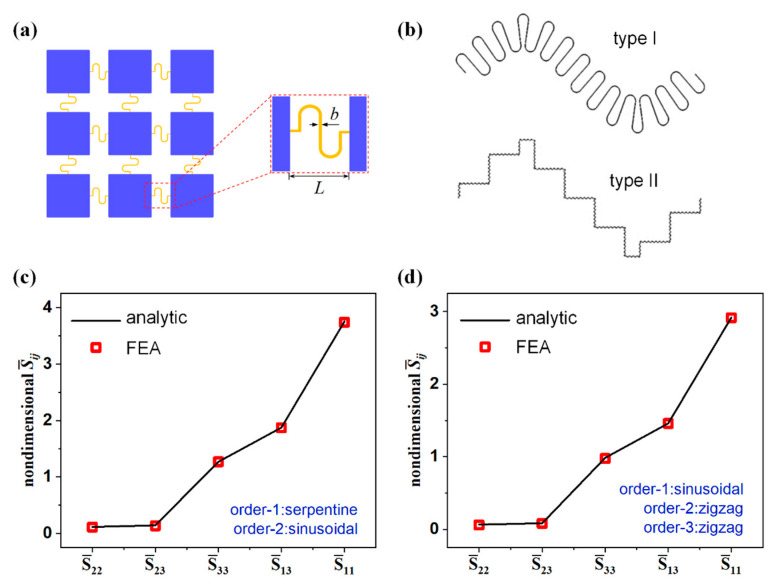
(**a**) Schematic of the “island–bridge” structure and illustration of the geometric parameters. (**b**) Two types of interconnects with two and three orders. Each component of the dimensionless flexibility matrix is calculated by the hierarchical theory and the FEA for (**c**) the type I and (**d**) type II interconnects.

**Figure 6 nanomaterials-13-02542-f006:**
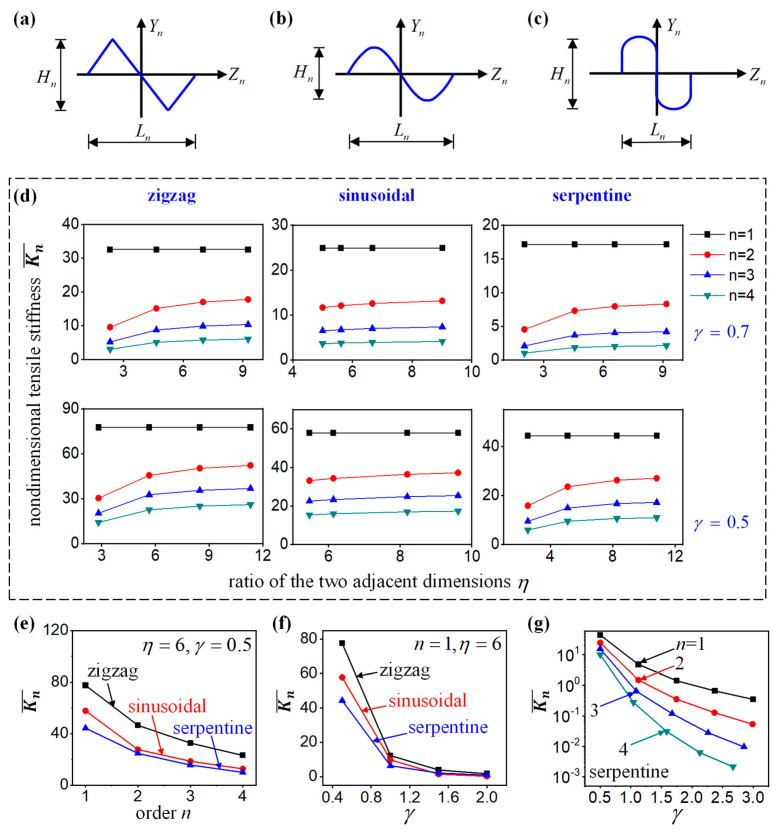
The self-similar interconnects: (**a**) The zigzag interconnects, (**b**) the sinusoidal interconnects, and (**c**) the serpentine interconnects. (**d**) The dimensional tensile stiffness ${\bar{K}}_{n}$ of the self-similar interconnects versus the ratio of the two adjacent dimensions η with different orders and height/span ratio. (**e**) The relationship between dimensionless tensile stiffness and the number of fractal orders for the zigzag, sinusoidal, and serpentine fractal-inspired interconnects with η=6 and γ = 0.5. (**f**) The relationship between dimensionless tensile stiffness and the height/span ratio γ for the zigzag, sinusoidal, and serpentine fractal-inspired interconnects with η=6 and *n* = 1. (**g**) The relationship between dimensionless tensile stiffness and the height/span ratio γ for the serpentine fractal-inspired interconnects with η=6 and different fractal orders.

## Data Availability

Not applicable.

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
