# Peer review of "A Hierarchical Theory for the Tensile Stiffness of Non-Buckling Fractal-Inspired Interconnects"

_nanomaterials, 2023, doi:10.3390/nano13182542_

Round 1
Reviewer 1 Report
The manuscript is academically well written although the language needs improvement. The reason for carrying out the work was well justified and I believe the developed hierarchical theory will be useful for stretchable inorganic electronics. The manuscript could be published after correcting the language errors in the manuscript.
There are several grammatical mistakes in the manuscript so they need to be corrected before publication
Reviewer 2 Report
The article is well prepared and has a large amount of theoretical knowledge. I have only one comment regarding a small review of the literature from the last 2 years. Literature research should be expanded
Reviewer 3 Report
This paper discusses the challenges associated with increased tensile stiffness introducing the fractal design into the non-buckling stretchable interconnects and presents a hierarchical theory for the tensile stiffness with arbitrary shape. It is also verified with the finite element analysis. Although the paper is well structured and provides a comprehensive overview of the topic, it seems to lack novelty. Many other papers already reported about the stiffness and even strain distribution using many strain-releasing structures including buckling, serpentine, zig-zag and so on. The authors claimed the novelty of the paper lies in introducing the hierarchical theory, however, the necessity of this method is not very convincing. I recommend that this paper be rejected, with the potential to resubmit after substantial changes.
1) Please clearly explain why and in what way this method has novelty, compared to the other papers (ex. 10.1016/j.ijsolstr.2014.07.025., 10.1002/adfm.201702589, 10.1016/j.actamat.2013.09.020., 10.1016/j.jmps.2019.01.019, 10.1115/1.4035118, 10.1088/2631-8695/ace2ab etc.)
